# Yeast Sup35 Prion Structure: Two Types, Four Parts, Many Variants

**DOI:** 10.3390/ijms20112633

**Published:** 2019-05-29

**Authors:** Alexander A. Dergalev, Alexander I. Alexandrov, Roman I. Ivannikov, Michael D. Ter-Avanesyan, Vitaly V. Kushnirov

**Affiliations:** A.N. Bach Institute of Biochemistry, Federal Research Center “Fundamentals of Biotechnology” of the Russian Academy of Sciences, Moscow 119071, Russia; alexanderdergalioff@gmail.com (A.A.D.); alexvir@gmail.com (A.I.A.); ivannikov_ra@mail.ru (R.I.I.)

**Keywords:** prion, amyloid, prion variants, prion core, proteinase K, Sup35, Rnq1

## Abstract

The yeast [*PSI*^+^] prion, formed by the Sup35 (eRF3) protein, has multiple structural variants differing in the strength of nonsense suppressor phenotype. Structure of [*PSI*^+^] and its variation are characterized poorly. Here, we mapped Sup35 amyloid cores of 26 [*PSI*^+^] ex vivo prions of different origin using proteinase K digestion and mass spectrometric identification of resistant peptides. In all [*PSI*^+^] variants the Sup35 amino acid residues 2–32 were fully resistant and the region up to residue 72 was partially resistant. Proteinase K-resistant structures were also found within regions 73–124, 125–153, and 154–221, but their presence differed between [*PSI*^+^] isolates. Two distinct digestion patterns were observed for region 2–72, which always correlated with the “strong” and “weak” [*PSI*^+^] nonsense suppressor phenotypes. Also, all [*PSI*^+^] with a weak pattern were eliminated by multicopy *HSP104* gene and were not toxic when combined with multicopy *SUP35*. [*PSI*^+^] with a strong pattern showed opposite properties, being resistant to multicopy *HSP104* and lethal with multicopy *SUP35*. Thus, Sup35 prion cores can be composed of up to four elements. [*PSI*^+^] variants can be divided into two classes reliably distinguishable basing on structure of the first element and the described assays.

## 1. Introduction

With few exceptions, prions represent an infectious variety of amyloids. Amyloids are fibers composed of many molecules of the same protein (protomers), whose altered conformation is reproduced along the fiber. Thus, amyloids can be considered as one-dimensional crystals, in contrast to more common 3D crystals. In amyloid fibers, polypeptide chains are perpendicular to the fiber axis and form multimolecular beta sheets, which are parallel to the axis. However, finer details of amyloid structure are poorly characterized. In many cases, and in particular in yeast prions, amyloids follow two novel and unprecedented structural principles: each protomer forms a single 4.7Å layer along the fiber axis, and neighboring polypeptide chains are located parallel and in-register [1,2]. Usually only a part of a protomer adopts amyloid structure and forms a stem of amyloid, while the rest of protomer is loosely attached to this stem [3] and can even retain its original structure and function [4]. Such division is especially clear in yeast prionogenic proteins, which contain separate functional and aggregation domains.

In yeast, prions manifest themselves as heritable genetic elements. The best studied of yeast prions is [*PSI*^+^], formed by the prion form of the essential translation termination factor eRF3, also known as Sup35. Sup35 is composed of three domains [5,6]. The N-terminal (N) domain (amino acid residues 1–123) is necessary and sufficient for the [*PSI*^+^] propagation. The middle (M) domain (residues 124–253) interacts with the Hsp104 chaperone through the region 128–148, and is also involved in the liquid-liquid phase separation behavior of Sup35 [7,8]. The essential C-terminal domain (residues 254–685) acts in translation termination [6]. 

A single protein can form amyloids with different structures. In the case of yeast Sup35 protein these structures manifest as [*PSI*^+^] variants differing in nonsense suppression and mitotic stability [9]. The relation between prion phenotype and amyloid structure is mediated, in particular, by quantitatively different activity of the Hsp104 chaperone, assisted by the Hsp70 and Hsp40 chaperones, towards different prions [10]. Hsp104 severs Sup35 amyloids into smaller pieces, thus multiplying amyloid ends, which can engage new Sup35 molecules into amyloid [11,12]. The amyloid structure defines how frequently it is recognized by Hsp104 and/or how easily Hsp104 can extract a protomer, thus cleaving an amyloid particle in two pieces. More frequent cleavage usually results in smaller, but more numerous, amyloid particles, faster amyloid conversion, lower levels of nonaggregated functional Sup35, higher suppression, and more stable inheritance [10]. Such [*PSI*^+^] are called “strong”. Less frequent Hsp104 action results in “weak” [*PSI*^+^], distinguished by moderate suppression and higher levels of nonaggregated Sup35. The actual number of alternative Sup35 amyloid structures, manifested as [*PSI*^+^] variants, is unclear and can be very high, but usually not more than four [*PSI*^+^] variants can be reliably distinguished by nonsense suppression [13] or other phenotypes [14,15,16]. It is also unclear whether all [*PSI*^+^] variants can be divided into two distinct groups, such as strong and weak, or they represent a continuum with gradually altering phenotype. Most studies tested only one representative of each type, which did not allow addressing this question. However, measurement of nonsense readthrough in 51 fresh [*PSI*^+^] isolates [17] indicated in favor of the latter opportunity.

The Sup35 prion structure was studied in several works, but it is still far from being fully understood and the proposed structures are somewhat contradictory [18,19,20]. 

A powerful, though rarely used approach for such studies is mapping of areas resistant to digestion with proteinase K (PK) (termed the prion core) using mass spectrometric identification of digestion-resistant peptides. Previous studies of the Sup35 prion structure with PK used Sup35NM amyloids obtained in vitro at either 4 or 37 °C, which mainly produce, respectively, strong and weak [*PSI*^+^], when introduced into yeast cells. However, it is unclear, how faithfully such amyloids recapitulate the properties of prions existing in vivo. In particular, such amyloids are to a certain extent heterogeneous [21] and this is difficult to control. 

Here, we developed a set of methods to isolate prions from yeast cells with sufficient quality and analyze their structure using PK digestion. This allowed making precise maps of the Sup35 prion core of many [*PSI*^+^] variants and correlate digestion patterns with [*PSI*^+^] phenotypes. Strikingly, our results differed considerably from those obtained with in vitro fibrils [22].

## 2. Results

### 2.1. General Structure of Sup35 Prion

To characterize the Sup35 amyloid structures in different [*PSI*^+^] variants, we mapped the Sup35 regions resistant to PK, which likely correspond to amyloid core regions. PK-resistant peptides were identified by matrix-assisted laser desorption/ionization-time-of-flight mass spectrometry (MALDI-TOF). The studied [*PSI*^+^] variants were existing and new isolates obtained in yeast strains 5V-H19 and 74-D694 in standard and nonstandard ways (Materials and Methods). For clarity, all [*PSI*^+^] isolates were named so that the first letter indicated the [*PSI*^+^] type, as defined in this work: S, strong; or W, weak. 

Sup35 prions were isolated from yeast cells overproducing Sup35 N and M domains (residues 1–239) fused to hexahistidine and GFP (Sup35NMG) under control of a *GAL1* promoter, and encoded on a multicopy plasmid. The Sup35 prion samples were isolated and digested with PK in the presence of 0.5% Sarcosyl. This ionic detergent does not inhibit PK activity and does not disassemble amyloid fibers. Notably, in contrast to earlier reports that PK cuts proteins after aliphatic and hydrophobic residues [23] and public databases (https://web.expasy.org /peptide_cutter/), we did not observe any significant residue specificity. Almost all types of amino acid residues were found at the C-termini of identified Sup35 fragments, except for proline. 

As predicted [24], the N-terminal methionine of Sup35 was fully removed and replaced with an acetyl group. 

PK-resistant peptides were found along the whole length of the Sup35 N and M domains, up to residue 221. Their location was not random: in all [*PSI*^+^] variants they were enclosed within regions 1–72, 73–124, 125–153, and 154–221, termed Regions 1, 2, 3, and 4 in this work (Figure 1). The corresponding PK-resistant structures in these regions were denoted as Cores 1, 2, 3, and 4. A small exception was a rarely observed group of fragments spanning Regions 2 and 3. Core 1 was present in all [*PSI*^+^] isolates, while the presence of Cores 2 to 4 varied. Region 222–233 was never protected from PK. The remaining Sup35 region 234–239 was often found as a part of PK-resistant peptides, but the other part of all of these peptides represented sequences unrelated to Sup35, such as hexahistidine and GFP, and so these peptides were ignored. 

No PK-protected fragments of Sup35NM were observed when Sup35NMG (preparations W-T7, S-W8-R41, and S-VH0) was boiled prior to PK digestion, which should dissolve all structures. This indicated that PK fully digested unstructured Sup35NM, and so all Sup35NM PK-resistant fragments observed in [*PSI*^+^] preparations reflected structures within Sup35.

PK digestion of soluble Sup35NMG isolated from 74-D694 [*psi*^−^] ∆*rnq1* cells using hexahistidine tag revealed no peptides from Regions 1–3 and only a minor amount of peptides belonging to Region 4. Thus, soluble Sup35NM lacks PK-resistant structure, except for a small fraction of molecules structured in the Region 4.

In earlier works, the mass spectral (MS) analysis of PK digestion was represented by simply showing MS spectra [21,22]. However, in this work such an approach was convenient only to represent digestion of Core 1, but not the whole Sup35NM profile, because the number of identified PK-resistant peptides was significantly higher than in previous works, approximately 30 to 60 peptides in each preparation, and often the peptides from different regions of Sup35NM were located in close proximity on an mass spectrum. For convenient presentation of these data, we developed a novel procedure. We summed up these peptides accounting to their apparent abundance as follows. For every residue of Sup35NM, we calculated the PK resistance index (R) as a sum of MS peak areas of PK resistant peptides, which include this residue. This index was normalized against its maximum value for each preparation, so that each graph would fit in the range of 0 to 1 (Figure 2A and Appendix A). 

Though MALDI is not regarded as a quantitative procedure, the obtained profiles were sufficiently reproducible. In particular, the profiles were very similar between weak [*PSI*^+^] variant W-1112 and its derivatives, and similar to W-Pb and W-HT2889 (Appendix A). For W-1112, we also tested whether PK resistance profile depends on Sup35NMG production level. Prion fractions were isolated from W-1112 cells without Sup35NMG production and with moderate overproduction from single-copy *GAL1* promoter. The obtained profiles were similar to the one obtained with high Sup35NMG overproduction from multicopy *GAL1* promoter (Figure 1A). This indicates that high overproduction did not significantly alter prion structure. However, analysis of preparations with no or low Sup35NMG overproduction was complicated by high background and presence of peptides unrelated to Sup35. For this reason, we used high Sup35NMG overproduction in this study.

### 2.2. C-Terminal Core of the Rnq1 Prion 

Two peptides, frequently found in Sup35NMG PK digests were identified as the Rnq1 fragments 366–405 and 360–405 (Appendix A). Ten more Rnq1 fragments were minor and found mostly in preparations where Sup35 was not overproduced. In mirror similarity to the Sup35 Core 1, all these fragments ended at the C-terminus. The size of Rnq1 peptides was from 26 to 50 residues. No Rnq1 peptides were observed in preparations from [*pin*^-^] and *Δrnq1* cells. 

### 2.3. Weak and Strong [PSI^+^] Differ in Core 1 Structure

While the major visual difference between [*PSI*^+^] isolates was the presence or absence of Cores 2, 3, and 4, the strength of the [*PSI*^+^] phenotype correlated with less prominent differences in the structure of Core 1, which are better seen on MS spectra (Figure 2B and Appendix A). We observed two major patterns of Core 1 digestion, which fully correlated with our initial assignment of [*PSI*^+^] variants as weak or strong basing on colony color and nonsense codon readthrough (Figure 3).

In all studied samples the N-terminal region of about 31 residues was fully protected from PK digestion. This region spanned residues 2–32 in weak variants and 2–32 or 2–30 in strong ones, but the 2–30 peptide was minor and thus poorly accessible. Partial protection from PK usually extended over a larger area, up to residue 72. 

In the strong [*PSI*^+^] variants the most abundant PK-resistant peptides were 2–35 and 2–38, while in weak variants these were 2–42 and 2–45. Another significant difference was that the region 42–72 was much better protected in strong [*PSI*^+^] variants, in which we observed less peptides ending in this region. As a result, a group of peptides 2–70, 2–71, and 2–72 was often the most abundant in strong [*PSI*^+^] preparations, while in weak [*PSI*^+^] it was minor or almost absent. The variant S-W8 was unique among strong [*PSI*^+^], since, in addition to characteristic peptides 2–35 and 2–38, it showed a major resistant peptide 2–57 instead of 2–70, thus somewhat resembling weak variants (Figure 2B and Appendix A). 

Interestingly, the PK resistance of region 33–72 depended on its connection to the fully protected core 2–32. Of all possible internal peptides of the region 33–72, we observed only trace amounts of peptides 38–70 and 38–71, and only in some strong [*PSI*^+^] preparations. This shows that the region 33–72 becomes highly sensitive to PK after being cleaved from the fully protected 2–32 core. 

Core 2 usually occupied region 91–120 in weak [*PSI*^+^] and 81–100 in strong [*PSI*^+^] (Table 1). In weak [*PSI*^+^] Core 2 was found more frequently and showed higher PK resistance. In S-W8 variant Core 2 was intermediate between the weak and strong types and occupied the region 91–113. Core 3 usually spanned the region 125–148 and was more frequent in strong [*PSI*^+^] isolates (Table 1). Cores 2 and 3 were less PK-resistant than Core 1 and their visible proportion decreased at higher levels of PK.

### 2.4. Additional Phenotypical Assays Distinguishing Weak and Strong [PSI^+^] Types 

Since our PK digestion data suggested that various types of [*PSI*^+^] belonged to two distinct structural classes, and did not represent a continuum of structural variants, we determined whether their difference could be observed as clearly in other assays. Earlier we and others observed that strong and weak [*PSI*^+^] react differently to overexpression of the *SUP35* and *HSP104* genes. Strong [*PSI*^+^] was extremely toxic or lethal in combination with multicopy *SUP35* [25] and was not cured by multicopy *HSP104* [12]. Weak [*PSI*^+^] variants showed opposite behavior, being nontoxic with multicopy *SUP35* [25] and very sensitive to multicopy *HSP104* [14,26]. However, the earlier studies tested single representatives of strong and weak [*PSI*^+^], and so it was not clear, how general these observations are. 

We transformed 15 [*PSI*^+^] variants with multicopy plasmids with the *HSP104*, *SUP35*, as well as *SUP35NM* genes (Figure 4). Multicopy *HSP104* eliminated all weak [*PSI*^+^], and did not eliminate strong ones. Multicopy *SUP35* was highly toxic for S-VH and S-W8, and lethal for other strong [*PSI*^+^]; it was slightly toxic for S-L2 and nontoxic for other weak [*PSI*^+^]. The effect of excess Sup35 on strong [*PSI*^+^] cells was so dramatic that even low copy centromeric *SUP35* (presumably, single-copy due to negative selection) caused significant retardation of growth of the S1 [*PSI*^+^] (Figure 4). Thus, all variants showed either one of the two described opposite modes of behavior in strict dependence on structure of the Core 1. Even the variant S-W8, which showed some structural similarity with weak [*PSI*^+^] and the nonsense readthrough level intermediate between strong and weak variants, was unambiguously classified as strong. Therefore, strong and weak [*PSI*^+^] variants represent two distinct classes, which are readily distinguished by the structure of Core 1 and by the effects of multicopy *SUP35* and *HSP104* genes.

The lethality of strong [*PSI*^+^] combined with multicopy *SUP35* was relieved when *SUP35* was supplemented with an equal number of copies of the *SUP45* gene. This confirms earlier reports that the lethality is related to depletion of the essential translation termination factor Sup45 (eRF1) [25]. Moreover, multicopy *SUP45* caused a significant antisuppressor effect, as observed earlier [27]. Multicopy *SUP35NMG* was less toxic than *SUP35*. This agrees with the Sup45 depletion model, since Sup35NMG lacks the Sup35 C domain, which binds Sup45. 

Our data on [*PSI*^+^] curing by *HSP104* overproduction driven by its endogenous promoter may appear to contradict earlier observations that low copy *HSP104* under strong inducible *GAL1* promoter cured strong [*PSI*^+^] with intermediate efficiency [9,26]. To resolve this contradiction, we transformed cells bearing S1 and S-7 variants with multicopy *HSP104* under its endogenous promoter and plated cells to SC-glucose and SC-galactose media. Transformants were streaked to single cells on YPD-red medium. No red [*psi*^−^] colonies were observed after SC-glucose, while these constituted ~20% of colonies after SC-galactose (Appendix A). Thus, [*PSI*^+^] curing by Hsp104 overproduction depends on carbon source, which resolves the contradiction. 

### 2.5. Manipulations with Core 2

To assess whether Core 2 could be selectively eliminated and whether this would affect the nonsense suppressor phenotype, we overproduced the Sup35 deletion variant Sup35(∆84–112)NMG, lacking a part of Region 2 in [*PSI*^+^] variants S1, S-W8 and W-1112. We reasoned that the truncated Sup35 should efficiently incorporate into Sup35 prion polymers via Core 1 [13,15,28], but would not template Core 2, which should then be eliminated. Then the plasmid encoding Sup35(∆84–112)NMG was lost, Sup35NMG was overproduced, and prion structures were analyzed by PK. Core 2 disappeared in variant S1, but stayed unchanged in W-1112 (Figure 5A). In S-W8, Core 2 remained, but was altered, and Core 3 appeared in one of two cases (Figure 5A). This suggests that Core 1 can predetermine the formation and restoration of Core 2, though not in all cases. 

To further characterize the dependence of Core 2 on Core 1, we inserted an additional copy of oligopeptide Repeats 4 and 5 between Regions 1 and 2 without altering these Regions. This protein—Sup35(N+2R)MG—was overproduced in variants W-1112 and S-W8, and its structure was analyzed. Core 2 was essentially unchanged in both variants (Figure 5B). Thus, in some contrast with the previous observation, Core 2 can propagate relatively independently from Core 1. The distance between Cores 1 and 2 was not critical for Core 2 formation, which suggests that the PK-sensitive region between Cores 1 and 2 is poorly structured or unstructured.

Notably, loss of Core 2 in the S1 variant did not cause a statistically significant change of nonsense readthrough (Figure 3). In earlier experiments [13] we observed that shuffling of Sup35 for Sup35(∆84–112) and back did not change phenotype of both strong and weak variants. This suggests that such procedure either did not change Core 2 or that Core 2 did not affect phenotype.

### 2.6. Instability and Heterogeneity of [PSI^+^] Variants

While some [*PSI*^+^] variants resisted directed attempts to eliminate Core 2, in some others Cores 2 and 3 were unstable and could change spontaneously during propagation. We noted that in one of the transformants of S-7 variant with pYes2-Sup35NMG overproduction of Sup35NMG results in a significant proportion of elongated cells, which could indicate defects of cytoskeleton related to insufficient function of Sup35 or Sup45 [29]. This prompted us to compare structures of Sup35NMG isolated from this (S-7-3) and two other transformants (S-7 and S-7-4) with standard cell morphology. All three structures differed in either Core 2 or Core 3. Isolate S-7-3 was distinguished by the presence of Core 3 (Appendix A). However, it is not certain that this difference is related to phenotype. Also of note, all three transformants did not noticeably differ by colony color despite the differences in Cores 2 and 3 (Appendix A). 

Some preparations, e.g., S1, S-R4b, and W-T7, appeared to represent a mix of two types of Core 2, while S-L20 combined Core 2 and Core 2–3 (Appendix A). This probably indicates heterogeneity and instability of Core 2 in these variants. Another issue is that many preparations contained relatively small amounts of resistant peptides from Regions 2, 3, and 4. This could be due to two reasons: either these structures have low resistance to PK or these structures are present only in a small proportion of molecules. 

### 2.7. Involvement of the PK-Protected Regions in Intermolecular Structure 

All Sup35NMG preparations isolated from [*PSI*^+^] cells contained amyloid structure, since they all were insoluble in sarcosyl and bound thioflavin T. However, this does not mean that all Sup35NM PK-resistant Cores were involved in amyloid intermolecular interactions. 

The amyloidogenic potential of Regions 1 and 2 is usually not questioned. Both regions alone can form amyloid fibers in vitro, and these can transform yeast cells to the [*PSI*^+^] state [15,30]. In line with this, we observed in vivo Sup35 amyloids, in which the only PK-resistant core resided in either Region 1 or Region 2. The former were found in several [*PSI*^+^] variants, in particular in S-L6, S-Cl2, W2, and W-VL (Appendix A). The latter has never been observed for intact Sup35NMG, but it was obtained by overproduction of the Sup35NMG protein lacking the first 30 residues in the 74-D694 [*PIN*^+^] cells. This protein formed amyloid, though with decreased efficiency, and the only PK-protected region in this amyloid included residues 81 to 113 (Appendix A, Table 1). This allows assuming that Regions 1 and 2 are involved in the amyloid core in all cases when they are protected from PK. 

The nature of Cores 3 and 4 is less certain. Recently, Sup35 amyloid was obtained in vitro, in which the only PK-resistant core spanned region 81–148 and thus included Regions 2 and 3 [22]. However, amyloid structures based solely on Region 3 were never shown. The same is true for Region 4, which, besides, contains 52% of charged residues and thus appears unlikely to form amyloid. Our experiments (Appendix A) indicate in favor of intermolecular amyloid structure of Regions 1 to 3, but leave uncertainty about Region 4. However, more reliable data may be required for the final answer. 

## 3. Discussion

### 3.1. General Properties of the Sup35 Prion Core

The aim of this work was to establish through PK digestion the structure of Sup35 prion core, its possible variations, and their relation to phenotype. We found that the Sup35 prion core is composed of up to four PK-resistant elements, located within Regions 1 (residues 2–72), 2 (73–124), 3 (125–153), and 4 (154–221). Curiously, the border between the Regions 2 and 3 coincided well with that of the N and M domains. The N-terminal Core 1 was present in all preparations, while other Cores differed between variants (Appendix A). Core 1 was also the largest of these structures and best protected from PK. 

Core 1 mainly defined the [*PSI*^+^] nonsense suppressor phenotype, and so formation of Cores 2 to 4 was expected to have little effect on phenotype. For Core 2, this was confirmed in the case of S1 variant. Region 3 roughly corresponds to the sequence 129–148 involved in interaction with Hsp104 [7]. Deletion of this sequence changes colony color from white to pink [7], which resembles the difference between weak and strong [*PSI*^+^]. However, we observed that the difference in Core 3 did not affect colony color in derivatives of variant S7 (Appendix A). This suggests that the formation of Core 3 has little effect on interaction with Hsp104 and prion fragmentation by Hsp104 in particular.

Core 2 in variants W-1112 and S-W8 was highly stable and resisted attempts to remove it through transient overproduction of Sup35NMG with deletion overlapping with Core 2 or by insertion of two additional oligopeptide repeats between Cores 1 and 2. In contrast to this, in S-7, Cores 2 and 3 could change spontaneously. In variants S1, S-R4b, and W-T7, the PK-resistant peptides from Region 2 appeared to represent two overlapping cores, one of which was typical of strong, and one of weak [*PSI*^+^] (Appendix A). It also appears likely that low levels of PK-resistant peptides from Regions 2 and/or 3 in some variants are due to presence of Cores 2 and/or 3 in only a fraction of prion particles. Structural heterogeneity of Sup35 prion was observed earlier as “a cloud of variants” [31]. Also, electron microscopy of Sup35 preparations seeded in vitro by strong or weak [*PSI*^+^] lysates revealed that two types of Sup35NM fibrils—“thick” and “thin”—were simultaneously present in each preparation. In the thick fibrils all Sup35NM was folded, including the M domain, while in the thin fibrils a region approximately corresponding to the M domain was unfolded [32]. 

Our data support the amyloid nature of the Cores 1, 2 and 3, while the nature of Core 4 is questionable. While Region 4 contains a high number of positively and negatively charged residues and thus does not appear amyloidogenic, some earlier studies indirectly support the amyloid nature of Core 4. In the latter cited work [32] the folded state of the M domain was faithfully propagated along fibrils. It seems unlikely that such transmission could occur without formation of intermolecular amyloid structure. Another work showed that of seven leucine residues in the M domain at least five form in-register contact with another Sup35 molecule [2]. Since three of these leucines are located in Region 3, and four in Region 4, this indicates that both Cores 3 and 4 form intermolecular in-register contacts. 

### 3.2. Two Types of [PSI^+^]

[*PSI*^+^] variants are usually distinguished based on a quantitative parameter, the nonsense codon readthrough, and associated phenotypes such as colony color or growth rate on adenine omission media. By these properties, the variants are usually labeled as weak or strong, although there are more than just two observable grades of phenotype, and it was not clear whether all [*PSI*^+^] variants can be divided into two or more clearly distinguishable groups by phenotype and/or structure. Here, we established that indeed all studied [*PSI*^+^] variants can be divided into two reliably distinguishable groups, which can be called weak and strong [*PSI*^+^], and provided three qualitative criteria to distinguish them: PK digestion pattern of the Core 1 and reaction to the multicopy *SUP35* and *HSP104* genes. Of note, other ways of Hsp104 overproduction, in particular, Hsp104 production under *GAL1* promoter, may not distinguish strong and weak [*PSI*^+^] with the same clarity [9,26]. The mechanistic basis for the difference of two [*PSI*^+^] types remains to be established. As a likely possibility, it can be related to different modes of interaction of weak and strong Sup35 prions with chaperones. 

We tried to widen the variety of Sup35 prion structures by using nonstandard conditions for [*PSI*^+^] induction. S-Cl prions were obtained at 4 °C in the *Δrnq1* background; Pb and R4b variants were obtained using as inducer altered Sup35NMG proteins with insertion of 29 residues before the Region 1 or deletion of 29 residues from Region 2. Some [*PSI*^+^] were obtained through the stage of lethal prions similar to those observed by McGlinchey and coauthors [33]. However, all [*PSI*^+^] obtained in unusual ways showed properties typical of either strong or weak [*PSI*^+^] types. This suggests that the Sup35 prion structures not belonging to the two described [*PSI*^+^] types are either rare or do not exist. 

### 3.3. Comparison of the Sup35 Fibrils Obtained in Vitro and in Vivo 

Many previous studies used as a model of the Sup35 prion structure the Sup35NM amyloids spontaneously formed in vitro at 4 °C or at 37 °C, which are regarded as analogs of the strong and weak prion folds, respectively. PK digestion based mapping of such Sup35 amyloids was performed only recently [22], and some results were very different from ours. These authors observed that the major PK-protected peptides in the “strong” fibrils assembled at 4 °C were 2–38, –42, –45, –46, and 2–72 in the “weak” 37 °C assembly. This result is directly opposite to our observations based on 26 prions that the former pattern is typical of weak [*PSI*^+^], and the latter of the strong ones. (Figure 2, Appendix A). Another difference is that simultaneous presence of two or more of PK-resistant cores was not observed in the in vitro fibrils.

The same work described Sup35 amyloids, whose PK-resistant core did not include the N-terminal region, being located between residues 62 or 81 and 144 [22]. We observed similar PK-resistant fragments 81 and 91–144, but failed to find prions lacking Core 1, and it is doubtful whether such amyloids can propagate as prions in vivo for the following reasons. Such amyloids were formed by mutant protein Sup35NM(S17R), and their fold was transferable in vitro to wild type Sup35NM. The S17R mutation impairs Core 1 formation, and so do two Sup35 N-terminal deletions, which we tested. The Sup35(∆2–30)NMG protein formed amyloid lacking Core 1 (Appendix A), but polymerized inefficiently, with only about 15% of this protein being aggregated despite high overproduction from multicopy *GAL1* promoter. In contrast, the Sup35(∆2–16)NMG protein, overproduced in [*PIN*^+^] cells, polymerized efficiently, but formed amyloid with shortened Core 1. Finally, the amyloids with 62–144 core produced unstable [*PSI*^+^], when introduced into yeast cells [22]. Thus, it appears likely that such Sup35NM fold rearranges in vivo and acquires Core 1. If so, it would be very interesting to find, why some amyloid folds existing in vitro are so disfavored in vivo. The described observations show that the data obtained with Sup35 fibrils spontaneously formed in vitro may differ significantly from those obtained with in vivo prions. 

### 3.4. Problems in Discrimination of [PSI^+^] Variants 

This work revealed some natural problems in discrimination of prion variants within each of the two major [*PSI*^+^] types. First is that the traditional discrimination based on the nonsense suppressor phenotype may incorrectly reflect the underlying prion variant, being affected by factors unrelated to Sup35 prion, such as duplication of chromosome I carrying the *ade1-14* gene [34] or [*PIN*^+^] and [*SWI*^+^] prions, which, in concert, cause the [*NSI^+^*] nonsense suppressor phenotype [35]. Such cases were frequent among the tested [*PSI*^+^] isolates with similar genetic background. S-7 and WS2 and S-W8 and W-1112 pairs showed similar colony color, but different nonsense readthrough; S1 and S-7 showed similar readthrough, but different colony color; W-VK and W-VL had redder colonies, but higher readthrough than W-1111 and W-1112 (Figure 3).

Another issue is that Cores 2 and 3 can be almost phenotypically silent and/or unstable. This raises the question whether their presence, at least in some cases, should be regarded as a [*PSI*^+^] variant difference. 

Some weak [*PSI*^+^] variants could be distinguished through different effects of overproduced Sup35 (growth retardation in W-L2, antisuppression in W-T7, Figure 4), though not many variants can be distinguished in this way. [*PSI*^+^] variants can also react differently to overproduction, depletion, or mutations of chaperones [14,36,37,38,39], reviewed in [40]. However, in most, though not all, cases this allows distinguishing only strong and weak [*PSI*^+^], rather than variants within each of these groups. Some of [*PSI*^+^] isolates obtained in the absence of Btn2, Cur1, Upf1,2,3, and Siw14 proteins, or at reduced activity of Hsp104 are cured by normal levels of these proteins, and thus represent independent variants (reviewed in [41]). Fine details of PK digestion of the region 30–72 could also serve for variants discrimination, but this requires further investigation.

Possibly the most universal instrument for discrimination of the [*PSI*^+^] variants could be testing of their ability to copolymerize with a set of N-terminal Sup35 mutants. This approach allowed to discriminate four variants studied here [15,16], while the total number of strains, distinguishable in this way, exceeds 20 [42]. Of note, this approach reflects only the structural differences in Region 1. 

### 3.5. Details of Sup35 Prion Structure

The structures of Sup35 prion cores were analyzed previously by different methods [18,20,43], and emerging descriptions differed noticeably, sometimes even within a single study [18]. Our data support some earlier findings and contradict others.

The most prominent structural feature in our study was the fully protected N-terminal region 2–32. Proline substitution and glycine insertion mutagenesis showed that this region is most critical for [*PSI*^+^] propagation [15,16]. Remarkably, such alterations, made just three residues away from the N-terminus, interfered with three [*PSI*^+^] variants of four and, regrettably, closer positions were not tested. The majority of mutations impairing [*PSI*^+^] propagation also mapped to the same region [44]. Finally, the Sup35 region 1–61, fused to GFP, was sufficient to faithfully transmit specific prion fold of variants S-VH, W-VK, W-VL, and S-W8, when this fusion protein was seeded by cell lysates and the resulting amyloids were reintroduced to yeast [15,16]. 

The 2–32 region of Sup35 amyloids was also the best protected from Hydrogen/Deuterium exchange [20]. In contrast, site-specific labeling with fluorescent molecules pyrene maleimide and acrylodan indicated that the region 1–20 is loosely structured and does not form intermolecular contacts [18]. The latter result could be an artifact, because fibers were formed from a mix of labeled and unlabeled Sup35 molecules, while prion formation is sensitive to even single residue differences in this region and the dyes used, acrylodan (225Da) and pyrene maleimide (297Da), are twice larger than an average amino acid residue. Summing up the above observations, the region 2–32 represents the key element of Sup35 prion structure for all studied [*PSI*^+^] variants.

The difference in folding of the weak and strong prions was observable as different PK cutting of the region 32–45 (Figure 2B, Appendix A). Notably, PK cut at the same sites in all preparations, but the proportions were different. PK never cut after Q33, N36, A39, Q40, P41, G43, or G44, and this cannot be related to residue specificity of PK, since all these residues, except for P, were good targets for PK at other locations of Sup35NM. Thus, the similarity of cutting sites probably reflects similar local folding of this region. We suggest that the key structural difference of the strong and weak folds is located within the region 2–32 inaccessible for PK. This difference modulates exposure of the region 32–45, causing the difference in accessibility of residues 35, 38, 42, and 45, which allows distinguishing weak and strong folds. 

The region 33–72 seems to differ significantly in structure from 2–32. It was protected from PK only partially, and, moreover, this protection ceased when PK cleaved this region from region 2–32. This appears as if fragment 33–72 was protected only due to its location on the surface of the fully protected core 2–32. Together with Core 2, such sequence of structures is reminiscent of the structure reported by Krishnan and Lindquist [18], where the regions called Head (20–35) and Tail (~80–110) are involved in intermolecular contacts, while the intermediate region 35–80, called Central Core, is not. Then, the Head would correspond to the fully protected region 2–32, and the Central Core to the partially protected region and PK-sensitive part of Region 2, while the Tail coincides very well with Core 2. Moreover, in strong [*PSI*^+^] Core 2 was shifted about 10 residues towards the N-terminus relatively to “weak” Core 2, and the same shift was observed for the Tail. The only, but significant, difference is that Core 2 was optional, while the Tail was presumed to be of key importance for the prion structure. Of note, the region between the fully protected Core 1 and “weak” Core 2 is mainly formed by oligopeptide repeats (residues 41–97, Figure 1).

Gorkovskiy and coauthors mapped residues involved in in-register contacts between Sup35 molecules in the region 9–101 [19]. According to this work, Tyr-29 is not involved in intermolecular in-register contacts, and is presumed to be in a loop or turn, while Tyr-35 was presumed to form contacts, being in the middle of beta strand. However, in our hands Tyr-29 was not accessible to PK while Tyr-35 was. On the other hand, the finding that Gln-71 forms intermolecular contacts, while Tyr-73 does not [19], fits well with the border of the partially protected region at Gln-72. 

The “serpentine” model of the Sup35 prion structure proposes that it is composed of short (5–8 residues) beta strands and beta turns alternating in the serpentine manner [45]. Our data are poorly compatible with this model: First, because in the serpentine structure all parts of amyloid core are similar in their properties, while we observed asymmetry and fundamental difference of regions 2–32 and 33–72. Second, the N-terminal residues were fully protected from PK, which would be difficult to achieve at the edge of serpentine structure, and suggests that these residues are buried inside of the amyloid structure. Notably, the Rnq1 prion (this work) and fibers of anchorless PrP generated in vitro [46] showed a similar property: the terminal region was fully protected from PK, though in these cases it was C-terminus. Third, one can expect from the serpentine structure periodic accessibility to PK corresponding to serpentine turns, which was not observed. Fourth, Core 1 ends at the third of five oligopeptide repeats, while in the serpentine model it is unclear why the remaining two repeats would not realize their similar potential to form amyloid structure. 

Several observations suggest that Core 1 is larger in weak variants. The region protected from Hydrogen/Deuterium exchange spans residues 1–37 in strong and 1–70 in weak Sup35 fibers [20], which agrees well with much higher fragility of strong fibers [43] and their lower thermal stability [47]. In contrast, we observed that the region 33–70 was much better protected from PK in strong [*PSI*^+^] variants (Figure 2 and Appendix A). Other observations more consistent with ours are that Gly58Asp (*PNM2*) and Gly54Glu mutations in this region interfere with propagation of strong, but not weak [*PSI*^+^] variants [48,49]. Thus, despite the obvious discrepancy, these observations do not directly contradict each other, and may be related to different methods of observation.

The in vitro-obtained “weak” and “strong” Sup35 fibrils showed dramatic difference in fragility [43]. Explaining this can be helped by our observation that weak prions usually included Core 2, while strong prions usually lacked it. Due to this, weak fibrils should be wider, while fragility is inversely related to the second power of the width of elongated objects. 

The findings described in this work pose some further questions. Why is Core 1 much more important than Cores 2 and 3 in vivo, despite their comparable abilities to form amyloid in vitro? Why there are just two types of [*PSI*^+^] and these are so different in their properties? Answering these questions is important for understanding the nature of prions and their interaction with cellular machinery.

## 4. Materials and Methods 

### 4.1. Yeast Strains and Media

Yeast strains used were 74-D694 (*MATa ade1-14 ura3-52 leu2-3,112 his3-Δ200 trp1-289*) and 5V-H19 (*MATa ade2-1 ura3-52 leu2-3,112 SUQ5 can1-100*) and their derivatives harboring different variants of [*PIN*^+^] and [*PSI*^+^]. Strains 74-D694 [*PIN^+^*] (also called OT60 [50]) and 74-D694 [*PIN^+^*]_very high_ [51] were used for obtaining new [*PSI*^+^] variants, and a number of previously selected [*PSI*^+^] variants of 74-D694 and 5V-H19 strains were also taken for this work (Table 2).

Synthetic complete (SC) media contained 6.7 g/L yeast nitrogen base, 20 g/L glucose or galactose, and required amino acids. For colony color development, SC media contained reduced amount of adenine (7mg/L, or 1/3 of standard). Adenine-limiting rich medium (YPDred) contained 5 g/L yeast extract, 20 g/L peptone, 20 g/L glucose and 20 g/L agar. 

### 4.2. Selection and Characterization of [PSI^+^] Isolates

74-D694 [*psi^−^*][*PIN^+^*] cells were transformed with pCM190-SUP35NMG plasmid [53]. Sup35NMG was overproduced for ~24 h by growing these cells in liquid SC medium lacking doxycycline and uracil, and then cells were streaked on SC plates lacking adenine and supplemented with 20 µg/mL doxycycline. After ~10 days of incubation at 30 °C, Ade+ clones were picked up, and the presence of [*PSI*^+^] prion was confirmed by growing cells on YPD_red_ plates with 3mM guanidine-HCl. Isolates S1, WS2, W2, and W3 were obtained in this way, obtaining of other [*PSI*^+^] isolates is described in the text.

### 4.3. [PSI^+^] Variants Obtained in Nonstandard Ways

To widen the variety of Sup35 prion structures, we tried to obtain [*PSI*^+^] in various nonstandard ways. The transformants of 74-D694 *Δrnq1* strain with a plasmid pYes2-SUP35NMG were left on SC-Ura glucose plate at 4°C for approximately one month. Then, white microcolonies appeared on the surface of old red colonies. These white cells had Ade+ phenotype sensitive to GuHCl, and thus carried [*PSI*^+^]. Presumably, when glucose was used up, the plasmid-borne *GAL1*-*SUP35NMG* gene was derepressed and Sup35NMG was produced at some intermediate level. This caused [*PSI*^+^] appearance despite the lack of Rnq1, and the [*PSI*^+^] cells had a preference in growth due to Ade+ phenotype. All of about 30 such isolates had white colony color, which suggests that they all were of the strong type.

Some [*PSI*^+^] variants were obtained using altered Sup35 as inducer. 29 N-terminal residues from Sup35 of the yeast *Pichia methanolica* [54] were placed ahead of the *S. cerevisiae* Sup35 sequence. This protein does not co-aggregate with preexisting [*PSI*^+^] (to be published elsewhere), so we expected to obtain a variant with unusual or missing Core 1. Overproduction of this protein caused appearance of [*PSI*^+^] variants, two of which, S-Pb and W-Pb were analyzed and did not show unusual traits.

Two [*PSI*^+^] isolates were induced by overproduction of Sup35NMG protein with deletion of residues 84-112. This deletion overlaps with Core 2 and was expected to interfere with its appearance. However, both studied variants S-R4b and W-R4b contained Core 2 structures.

Some variants were obtained as nontoxic descendants of lethal or toxic [*PSI*^+^]. Such [*PSI*^+^] were obtained in the presence of the centromeric rescue plasmid pRS315 (*LEU2*) carrying *SUP35C* and *SUP45* genes and were unable to lose this plasmid, or lost it very infrequently. The rescue plasmid was lost from the studied descendants. The obtaining and analysis of such [*PSI*^+^] will be described elsewhere.

### 4.4. Plasmids

Plasmids producing the Sup35NMG protein were based on either pYES2 (Thermo Fisher Scientific, Waltham, MA, USA) or pCM190 [53] episomal vectors with *URA3* marker. DNA fragment encoding Sup35NMG was inserted between *Bam*HI and *Xba*I sites of pYES2 or into *Bam*HI site of pCM190. The Sup35NMG sequence expressed by these plasmids was Sup35(1-239)-6*His-Pro-Val-Ala-Thr-eGFP. Integrative variant of this plasmid was obtained by deleting *Nru*I-*Nae*I fragment containing replicator from 2-micron plasmid. Integration was targeted into the *ura3-52* mutant gene. To construct pYES2-SUP35(∆84–112)NMG plasmid we amplified the *SUP35* region with such deletion from pR1-4 plasmid [28] with primers GAL-Sup35-Df: GGATCGGACTACTAGCAGCTGCCCACTAGCAACAATG and Su35R1: CCTTCTTGGTAGCATTGGC. This DNA fragment was inserted into pYES2-SUP35NMG plasmid cut with PvuII and BalI using quick-fusion cloning kit (no longer produced by Bimake, Houston, TX, USA) to replace the related sequence. Production of Sup35NMG and its variants was repressed by glucose and induced by galactose for pYES2-based plasmids, and repressed by 20 mg/L doxycycline or induced by its absence for pCM190-SUP35NMG. pRS315-SUP35C-SUP45 is a centromeric *LEU2* plasmid encoding the Sup35 C domain and Sup45 proteins at native levels.

### 4.5. Prion Isolation From Yeast

[*PSI*^+^] cells were transformed with multicopy plasmids overproducing Sup35NMG under control of *GAL1* promoter. Cells were grown in 150 mL of non-inducing synthetic medium to OD600 = 2.5, and then transferred to 300 mL of inducing medium and grown overnight. Preservation of the original prion fold was confirmed by the lack of intracellular fluorescent Sup35NMG rings and by correct colony color of grown cells. Cells were collected to 50 mL Falcon tubes (~3 g of wet pellet) and lysed by vigorous shaking with glass beads in TBS (30 mM Tris-HCl, pH 7.6; 150 mM NaCl) with 5 mM PMSF, 1mM dithiothreitol, and 20 µg/mL of RNAse A. The lysate including cell debris was transferred to two 2.3 mL Eppendorf tubes and spun at 20,000 g, 4 °C, for 30 min. The pellet was resuspended in TBS with 2 mM PMSF and spun again, resuspended in 1.5 mL of TBS with 2 mM PMSF. Sarcosyl was added to 5%, and the mix was sonicated for 30 (6 × 5) s at 50% of power of VCX130 sonicator with a 2 mm tip (Sonics and Materials, Newtown, CT, USA). Lysate was centrifuged at 20,000 g, 4 °C, for 3 min. Supernatant (2.7 mL) was loaded to 3.5 mL open-top tube (Beckman Coulter cat. 349622) on top of a sucrose gradient made of 250 μL each of 60%, 40%, and 20% sucrose. Such “sucrose trap” allows removing a significant amount of nonprotein material, which goes to pellet. This material presumably represents polysaccharides, which otherwise interfere with some downstream procedures and decrease the quality of mass spectra. Tubes were spun in an MLS-50 bucket rotor (Beckman Coulter, Brea, CA, USA) at 268,000 g for 4 h. Sup35NMG was visually identified by GFP fluorescence in 60% sucrose fraction and collected by pipette.

### 4.6. PK Digestion and Mass Spectrometry

Sup35NMG (200 μg/mL) was digested by PK (25 µg/mL) in 20 µL for 1 h at room temperature. PK was inactivated by adding 1 µL of 100 mM PMSF and peptides precipitated by addition of 16 µL of acetone and incubation on ice for 10 min (0.5 mL PCR tubes were used). Peptides were collected by centrifugation at 16,000 g for 1 min, dissolved in 20 μL of water, precipitated with 30 μL acetone, dissolved in 15 µL of water, denatured by boiling for 3 min, and analyzed by MALDI-TOF/TOF mass spectrometer UltrafleXtreme (Bruker, Germany). Peptides were identified by tandem mass spectroscopy (MS-MS) and/or as groups of related peaks.

### 4.7. Nonsense Readthrough Measurement

Nonsense readthrough levels were measured for the UGA codon followed by cytidine (UGAc). This extended codon showed the highest readthrough among other codons in the S-7 variant of 74-D694 strain [55] and it should not be affected by the *SUQ5* UAA suppressor tRNA of 5V-H19. Yeast cells were transformed with pDB691 plasmid carrying tandem Renilla and firefly luciferase genes separated by the UGAc codon or with control plasmid pDB690 lacking intermediate nonsense codon [56]. The transformants were assayed with the dual luciferase reporter assay system (Promega, Madison, WI, USA), using a Glomax 20/20 luminometer (Promega, Madison, WI, USA). All assays were repeated three times and the data are expressed as mean ± SEM.

### 4.8. PK Resistance Index Calculation

To graphically represent the PK resistant structures, we calculated for every residue of Sup35NM the PK resistance index R as a sum of mass spectral peak areas of peptides which include this residue (Figure 2A and Appendix A). For each preparation, two MALDI spectra were taken: in the reflecton and in the linear mode. The former is more precise and convenient for identification of peptides, but underrepresents heavy peptides. The latter shows more correctly the amount of peptides in the range of 4 to 8 kDa. The peptides in the range of 1 to 4.5 kDa were calculated basing on reflecton spectra, for heavier peptides calculation was based on linear spectra with a correction coefficient allowing to merge two datasets by equalizing them at peptides 2–38 (4343.3 Da) or 2–42 (4710.7 Da). The resulting graphs were normalized against their maximum value for each preparation.

## Figures and Tables

**Figure 1 ijms-20-02633-f001:**
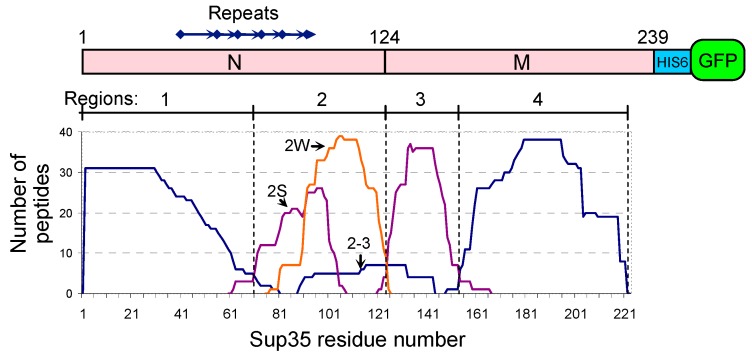
PK-resistant regions of Sup35 prions. A superposition of Sup35 peptides identified after PK digestion of Sup35NMG preparations from all [*PSI*^+^] isolates. The peptides were divided into six groups according to their start and end positions. Graph shows the number of identified peptides in each group, which include a given amino acid residue. In other words, the graph was obtained by placing identified peptides in accordance with their coordinates one above the other with a height of 1. Peptides of the groups 2S and 2W were found more often in Cores 2 of strong and weak [*PSI*^+^] variants, respectively. Peptides of the 2–3 group of fragments were observed in variants S-L20 and S-T9 at low abundance. Note that the large variation in the abundance and the frequency of appearance of the peptides is not reflected here.

**Figure 2 ijms-20-02633-f002:**
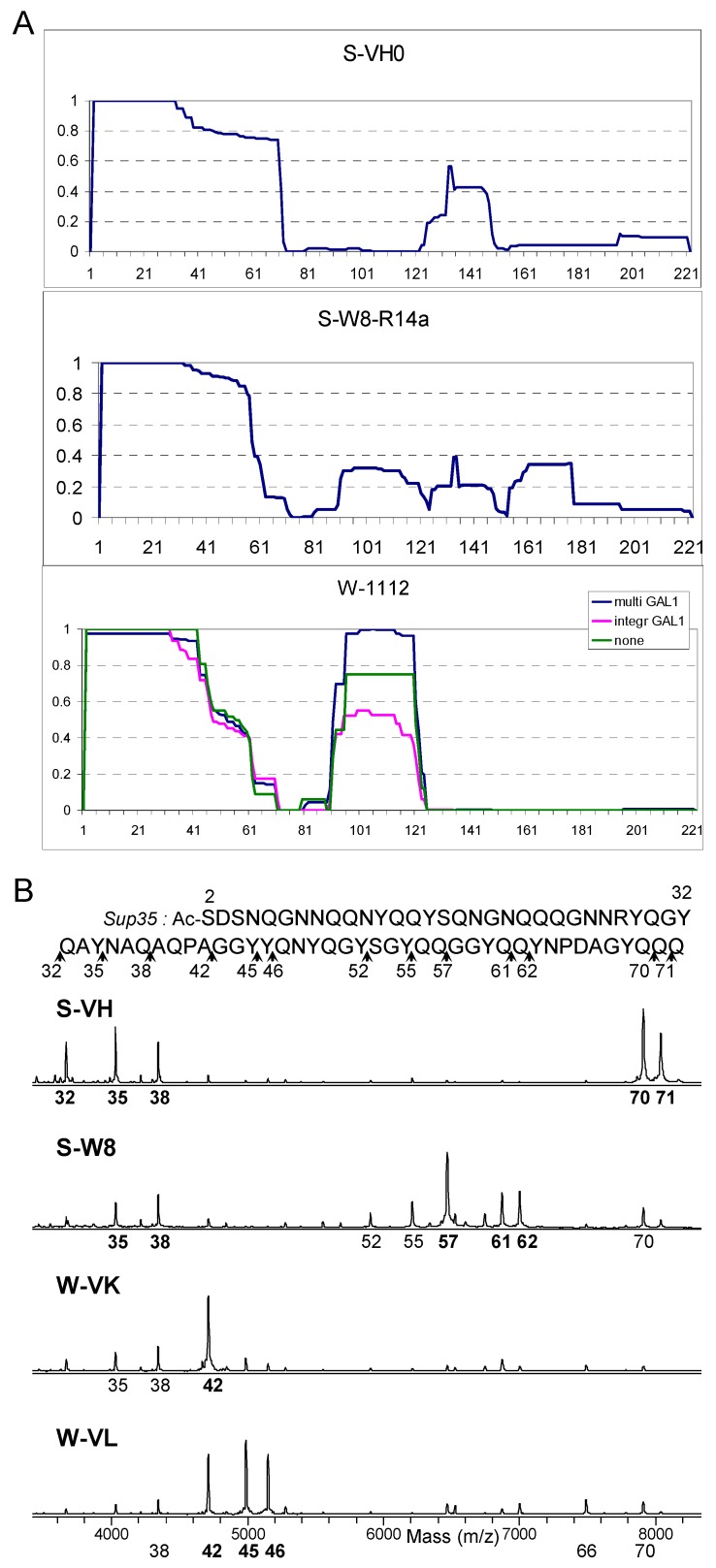
Structural analysis of selected [*PSI*^+^] variants. (**A**) PK resistance index for the Sup35 region 2–222 of variants L20, S-W8-R4, and W-1112, calculated as described in the text. In contrast to Figure 1, the contribution of each peptide was proportional to its MS peak area. (**B**) Matrix-assisted laser desorption/ionization-time-of-flight mass spectrometry (MALDI-TOF) spectra (linear mode) of PK digests of Sup35-NMG of S-VH, S-W8, W-VK, and W-VL variants. All peptides in this mass range in these spectra belong to Core 1 and start from Ser2, so only the number of their C-terminal residue is given below each spectrum. The variants in panel A were selected to represent the diversity of Core2 and Core 3 structures. Panel B shows different types of Core 1 on an example of a well characterized set of variants [15,16]. Other variants are presented in Appendix A.

**Figure 3 ijms-20-02633-f003:**
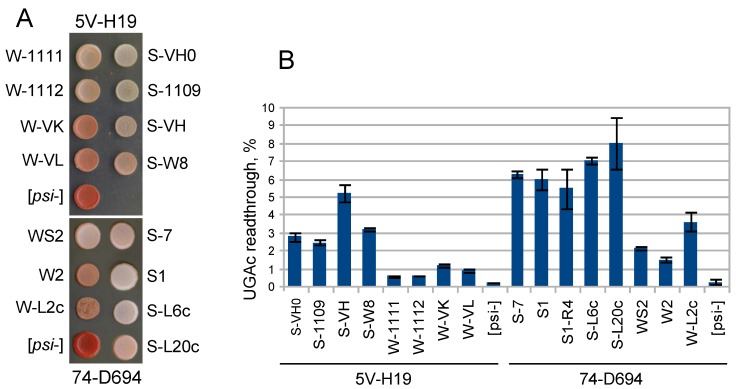
Suppressor phenotypes of [*PSI*^+^] variants. (**A**) Indicated variants were spotted to YPD-red plate and grown for 4 days. (**B**) Readthrough of the nonsense codon UGAc. Standard error is indicated.

**Figure 4 ijms-20-02633-f004:**
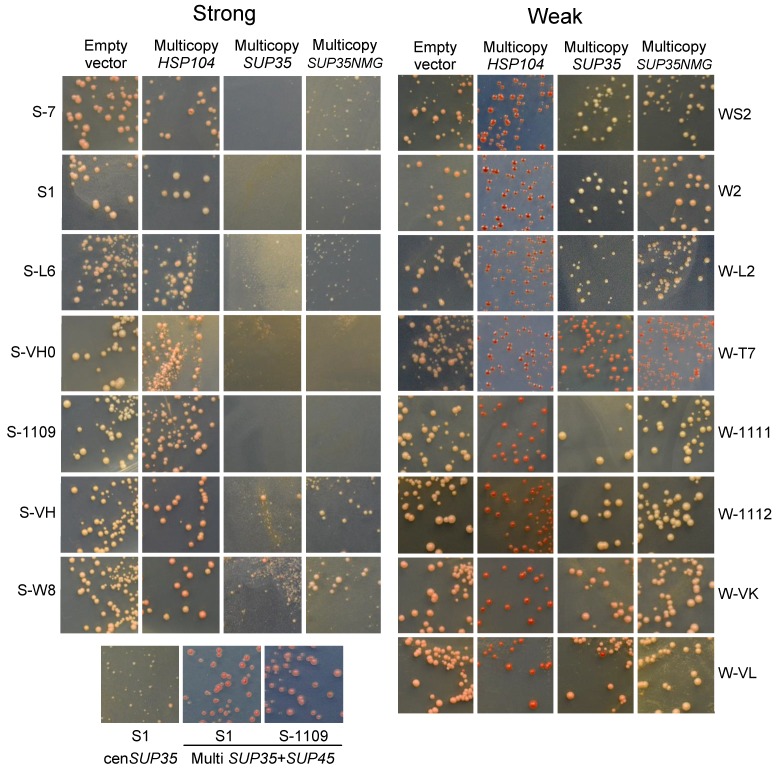
Strong and weak [*PSI*^+^] variants differ by the effects of overproduction of Sup35 and Hsp104. The indicated [*PSI*^+^] variants of 74-D694 and 5V-H19 strains were transformed with the multicopy plasmids based on YEplac181 (*LEU2*) carrying the *HSP104*, *SUP35*, and *SUP35NM* genes as indicated. Also, the “cen*SUP35*” panel used the centromeric pRS315-SUP35 *LEU2* plasmid and the “multi *SUP35*+*SUP45*” panels used multicopy YEplac195 *URA3* plasmid with *SUP35* and *SUP45* genes. SC media contained reduced amount of adenine for colony color development. Photographs were taken on the fourth day after transformation.

**Figure 5 ijms-20-02633-f005:**
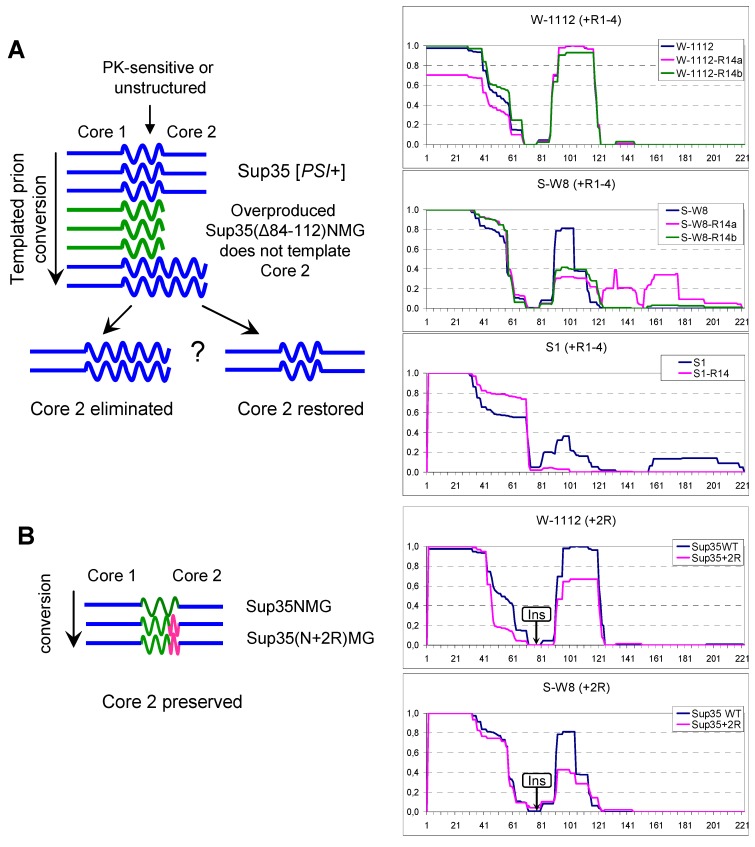
Manipulations with Core 2. Left: the schemes of experiments. Straight lines show amyloid Cores 1 and 2; wavy line: presumably unstructured regions; pink line: two added oligopeptide repeats (9+10 residues). Right: Sup35NM PK resistance profiles before (blue line) and after experiments. (**A**) Attempted Core 2 elimination by overproduction of Sup35(Δ84–112)NMG with deletion covering a larger part of Core 2. (**B**) Core 2 was propagated in variants W-1112 and S-W8 by Sup35N(+2R)MG with 2 repeats inserted between Cores 1 and 2. The region of two added repeats was protease-sensitive and it was omitted from the profile of Sup35N(+2R)M to allow comparison of the profiles; the place of insert is indicated as “Ins”.

**Table 1 ijms-20-02633-t001:** Location of Sup35 Cores 2 and 3.

Weak [*PSI*^+^]			Strong [*PSI*^+^]		
Name	Core 2	Core 3	Name	Core 2	Core 3
WS2	96,101–120 *	-	S-7	91–104	-
W2	-	-	S-7-3	81–100 *	125–148
W3	91–120	-	S-7-4	82,92–100,113 *	-
W-L2	(91,96,101)–123	133–153*	S1	82–100,92–113	-
W-T7	81–100,92–120	133–149	S1-R4	72,82–100 *	-
W-Pb	91–121	-	S-L4	72,82–100 *	-
W-R4b	(92,96)–(104,113,121)	133–148	S-L6	-	-
W-HT2889	91–122	-	S-L20	82–100	91–144*†
W-1111	91,96–120	-	S-Pb	92–115	-
W-1112	91–121	-	S-R4b	91–113	-
W-1112-R14a, R14b	91–121	-	S-Cl1	82,92–100,113 *	-
W-VK	96–(113,120)	-	S-Cl2	-	-
W-VL	-	-	S-VH0	-	125–148
			S-1109	82–100 *	-
	Amyloid		S-VH	82–100	-
Δ2-30	81–113	-	S-W8	91–(104,113)	-
			S-W8-R14a	91–121	(125,133)–148
			S-W8-R14b	91–121	-

The borders of the Cores were defined as the biggest increment or decrement of the PK resistance index R per three sequential residues. In cases of nearly equal maximal increments or decrements two or three values are given for borders. * Index R is below 0.1. † Core 2–3.

**Table 2 ijms-20-02633-t002:** The yeast strains and [*PSI*^+^] variants used.

Variant	Original Name/Comment	Source/Reference
	**Yeast Strain 74-D694**	
[*psi*^−^][*PIN*^+^]	OT60, parent for nontoxic [*PSI*^+^] obtained in this work	Y. Chernoff [50]
[*psi*^−^][*PIN*^+^]_very high_	Parent for the lethal and toxic isolates	S. Liebman [51]
WS2		This work
W2		This work
W3		This work
W-L2		This work
W-Pb		This work
W-T7		This work
W-R4		This work
S-7	7-74-D694, also known as OT56*	Y. Chernoff [50]
S1		This work
S1-R4		This work
S-L4		This work
S-L6		This work
S-L20		This work
S-Pb		This work
S-R4b		This work
S-Cl1		This work
S-Cl2		This work
W-HT2889	Heather True collection #2889	H. True
	**Yeast Strain 5V-H19**	
S-VH0†	Our collection; [*PSI*^+^] originates from Albert Hinnen strain AH216	[52]
S-1109	Our collection #1109	
W-1111	Our collection #1111	
W-1112	Our collection #1112	
S-VH†	VH	C.Y. King [15]
S-W8	W8	C.Y. King [16]
W-VK	VK	C.Y. King [15]
W-VL	VL	C.Y. King [15]

The [*PSI*^+^] variants were named so that the first letter indicates the [*PSI*^+^] type, as defined in this work; S: strong; W: weak. These letters were added to established names VH, W8, VK, and VL to avoid confusion, e.g., strong variant W8 was renamed to S-W8. *7-74-D694 and OT56 are strain names rather than names of a prion variant. †S-VH0 and S-VH are the same [*PSI*^+^] isolate, but they were separated to different labs for 20 years and [*PIN*^+^] was selectively cured with guanidine hydrochloride in S-VH [49].

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
