# Peer review of "Yeast Sup35 Prion Structure: Two Types, Four Parts, Many Variants"

_ijms, 2019, doi:10.3390/ijms20112633_

Round 1
Reviewer 1 Report
Comments for Dergalev et al
This is a nice study with important results for the field. It should be published pending some minor revisions (see below):
Line 12) Latin phrases are usually italicized elsewhere in the article so “ex vivo” should be italicized here
Line 115) remove the comma after “observed”
Line 165) Rotate image of Figure 3A 90 degrees so that it is the correct orientation. It can be reduced in size if need be but the current image seems optimized for space yet this is an online-only journal so print size is irrelevant. Images in figures should not require reader to rotate the page to read.
Line 198) Recommend revising “we looked whether” to “we determined whether”
Line 226) remove the comma after “relieved”
Line 238) the authors may want to mention this again in the discussion section as I think that this is significant finding given that many labs use GAL induction to study Hsp104 curing.
Line 248) Recommend revising “This allows thinking that” to “This suggests that”
Line 261) Revise “Core 2 or Core 2 did” to “Core 2, or that Core 2 did”
Line 269) Revise “Core3” to “Core 3”
Line 319-320) The mention that Sup35 residues 129-148 is involved in interaction with Hsp104 is unreferenced here (Helsen and Glover 2012??). What is unclear, but important to clarify, is that “interaction with Hsp104” can mean a variety of things. In this case I believe these residues were indicated as critical for Hsp104 curing (by overexpression), rather than residues necessary for productive interaction with Hsp104 for prion fragmentation. The authors should clarify.
Line 323-324) The claim that formation of core 3 has little effect on interaction with Hsp104 seems too broad, considering Hsp104 binding affinity assays or colocalization in vivo has not been shown. Instead, use softer language, like "Evidence suggests the formation of Core 3 may have little effect on interaction with Hsp104."
Line 357-358 and 405-406 relevant literature seems to be ignored/missing. Lines 357-358: "As a likely possibility, it can be related to different modes of interaction of weak and strong Sup35 prions with chaperones." This statement has no supporting evidence presented or citations but it brings up a big issue. Likewise on Lines 405-406: "Some [PSI+] variants could be distinguished... or chaperones.”—only one paper is cited (the authors own work from 2000). This paper was published nearly 2 decades ago and a tremendous amount of work has been done in that time on this subject. There is substantial evidence that chaperones interact differently with distinct prion variants and examples where variants can be distinguished by chaperone alterations. These topics were recently reviewed, broadly in Killian et al. Viruses 2019; and in the context of Hsp40s in Killian and Hines, Plos Pathogens, 2018. Both describe significant examples in the literature that are pertinent to the points raised by the authors (most notably see work by Stein and True, and Harris et al. 2014 discussed in both reviews). I strongly recommend the authors put these ideas in some context (with citations); as it reads now it sounds like these are just suggestions of things that might happen, but for which there is little evidence, which is far from the truth.
Line 447-448) "It was protected ... region 2-32." Revise sentence structure.
Author Response
Response to Reviewer 1 comments
The authors are very thankful to both Reviewers for good words about our work and the helpful suggestions and corrections. We agree with all comments and introduce the following corrections. We cite below only the Reviewer’s text which requires answer.
Line 12) Latin phrases are usually italicized elsewhere in the article so “ex vivo” should be italicized here.
Answer: Corrected
Line 115) remove the comma after “observed”.
Answer: Removed
Line 165) Rotate image of Figure 3A 90 degrees so that it is the correct orientation. It can be reduced in size if need be but the current image seems optimized for space yet this is an online-only journal so print size is irrelevant. Images in figures should not require reader to rotate the page to read.
Answer: Agree, Figure 3A has been rotated.
Line 198) Recommend revising “we looked whether” to “we determined whether”.
Answer: Corrected
Line 226) remove the comma after “relieved”.
Answer: Removed
Line 238) the authors may want to mention this again in the discussion section as I think that this is significant finding given that many labs use GAL induction to study Hsp104 curing.
Answer: We added to chapter 3.2: “Of note, other ways of Hsp104 overproduction, in particular, Hsp104 production under GAL1 promoter, may not distinguish strong and weak [PSI+] with the same clarity [9,26]”.
Line 248) Recommend revising “This allows thinking that” to “This suggests that”.
Answer: Corrected
Line 261) Revise “Core 2 or Core 2 did” to “Core 2, or that Core 2 did”.
Answer: Corrected
Line 269) Revise “Core3” to “Core 3”.
Answer: Corrected
Line 319-320) The mention that Sup35 residues 129-148 is involved in interaction with Hsp104 is unreferenced here (Helsen and Glover 2012??).
Answer: Agree. The reference (Helsen and Glover, 2012) was given in the next sentence, and now we added this reference to the sentence in question.
What is unclear, but important to clarify, is that “interaction with Hsp104” can mean a variety of things. In this case I believe these residues were indicated as critical for Hsp104 curing (by overexpression), rather than residues necessary for productive interaction with Hsp104 for prion fragmentation. The authors should clarify.
Answer: Clarification was required, though we meant the latter possibility, prion fragmentation. The sentence was corrected to: “This suggests that formation of Core 3 has little effect on interaction with Hsp104 and prion fragmentation by Hsp104 in particular.”
Line 323-324) The claim that formation of core 3 has little effect on interaction with Hsp104 seems too broad, considering Hsp104 binding affinity assays or colocalization in vivo has not been shown. Instead, use softer language, like "Evidence suggests the formation of Core 3 may have little effect on interaction with Hsp104."
Answer: Corrected, please see in the previous answer.
Line 357-358 and 405-406 relevant literature seems to be ignored/missing. Lines 357-358: "As a likely possibility, it can be related to different modes of interaction of weak and strong Sup35 prions with chaperones." This statement has no supporting evidence presented or citations but it brings up a big issue. Likewise on Lines 405-406: "Some [PSI+] variants could be distinguished... or chaperones.”—only one paper is cited (the authors own work from 2000). This paper was published nearly 2 decades ago and a tremendous amount of work has been done in that time on this subject. There is substantial evidence that chaperones interact differently with distinct prion variants and examples where variants can be distinguished by chaperone alterations. These topics were recently reviewed, broadly in Killian et al. Viruses 2019; and in the context of Hsp40s in Killian and Hines, Plos Pathogens, 2018. Both describe significant examples in the literature that are pertinent to the points raised by the authors (most notably see work by Stein and True, and Harris et al. 2014 discussed in both reviews). I strongly recommend the authors put these ideas in some context (with citations); as it reads now it sounds like these are just suggestions of things that might happen, but for which there is little evidence, which is far from the truth.
Answer: we introduced suggested citations and some more. However, we note that these works were mainly able to distinguish strong and weak [PSI+], while our focus was on discrimination of variants within each of these groups. The corrected text: “Some weak [PSI+] variants could be distinguished through different effects of overproduced Sup35 (growth retardation in W-L2, antisuppression in W-T7, Figure 4), though not many variants can be distinguished in this way. [PSI+] variants can also react differently to overproduction, depletion, or mutations of chaperones [14,36-39], reviewed in [40]. However, in most, though not all, cases this allows distinguishing only strong and weak [PSI+], rather than variants within each of these groups.”
Line 447-448) "It was protected ... region 2-32." Revise sentence structure.
Answer: We did not have a clear idea, what is wrong with this sentence. After trying several variants, we stopped on the following, and slightly extended the interpretation:
“The region 33-72 seems to differ significantly in structure from 2-32. It was protected from PK only partially, and, moreover, this protection ceased when PK cleaved this region from region 2-32. This appears as if fragment 33-72 was protected only due to its location on the surface of the fully protected core 2-32.”
Reviewer 2 Report
Review of “Yeast Sup35 prion structure: two types, four parts, many variants” by Dergalev et al.
In this work, the authors have extracted amyloid of Sup35 from a large series of [PSI+] variants. This amyloid was treated with proteinase K and the resistant peptides identified by mass spectrometry. The prion phenotype, sensitivity to curing by an intermediate level of Hsp104p overproduction and sensitivity to growth inhibition by overproduction of Sup35p were also measured for these variants. This represents an enormous amount of work in an area of central interest to the prion field.
1) This is an excellent paper, a big advance in the yeast prion area. In the past, the differences between strong and weak variants of the [PSI+] prion were determined by comparing one strong and one weak variant using various methodIs. As the authors point out (and it has been mentioned by others as well), this type of comparison does not prove that the physico-chemical difference between the variants was due to their being ‘strong’ or ‘weak’. Here, the authors examine a large group of variants, so one can see directly how representative are the structures and genetic properties that they examine.
2) Another serious shortcoming of past experiments was that the prion amyloid was made in vitro, and it was never clear what proportion of the amyloid was doing the infecting. True, the preparations were often/usually shown to be infections, but if 10% of the (potentially heterogeneous) filaments made in vitro were doing the infecting, most physical methods detect the properties of main mass of material and would be reporting on the other 90%. Here, the authors have isolated filaments from cells, currently the closest one can come to a homogeneous preparation.
3) While the in vivo material is the best one can do to make homogeneous amyloid, there are several papers in the literature showing that a prion constitutes a “cloud” of variants (to use Collinge’s expression), even in a single strain, mentioned by the authors (line 333). This work is important in being able to detect at least some level of physical heterogeneity, as discussed in that paragraph.
4) While isolation of filaments from in vivo has been done before, I think this is the first case where the purpose was for examination by MS. I am sure that other groups will want to use this method for examining variant differences that are not defined by ‘strong’ vs ‘weak’. For that reason, the authors should present the filament isolation method, the PK digestion method and the mass spec method in more detail. For example, how was the GFP-fluorescing fraction “collected” (lines 573-574). What kind of tubes were used for the acetone-precipitation of peptides and spinning, all in tiny volumes (lines 576-582)? What problems arose in the method, and how were they dealt with.
5) The authors assert that the [PSI+] variants examined fall nicely into the two classes, ‘strong’ and ‘weak’. But my view of the data suggests that the variants comprise a wide spectrum, with strong and weak representing the two ends in the dimension examined. For example, in Fig. 2A, the color spread seems like a continuous variation from near-white to very dark pink: S-W8 and S-VH (classified as strong) do not seem whiter than W-111 or W-112, all in strain 5V-H19. Likewise S-L6c and S-L20c look the same to me as WS2 in the background of strain 74-D694. While the readthru data in Fig 2B show nice separation of strong from weak groups, it is possible that other terminators or their surrounding sequences may not be so clear. The data in table 1 do show average differences between the strong and weak groups, but I don’t see any fragment that is generally present in one group and not found in the other. Lots of overlap and TREMENDOUS variability within each group.
In fact, a lot of work from many groups, including Kushnirov’s group, shows that yeast prion variants (including [PSI+] variants) can be separated by many different criteria, including lethality (without overproducing anything, many [PSI+] variants are severely toxic or lethal), sensitivity to overproduction or deficiency of various chaperones or other components, ability to overcome a transmission barrier based on Sup35p sequence differences (inter-species, intra-species or artificially created mutants), and other criteria as well. Many of these differences to not follow the strong vs weak paradigm, so it seems clear that the distribution of prion variants is multidimensional.
6) Lines 293 to 301 were a bit confusing to me. On first reading, I thought the authors were saying some variants had ONLY region 2 being PK – resistant. I think this is not what they intend, and a bit of re-phrasing might help.
7) Line 318, ”Core 1 mainly defined the phenotype…”. As mentioned in 5) above, there are other phenotypes of [PSI+] that do not correlate with strong vs weak and are not tested here. I suggest the authors make it, “Core 1 mainly defined the strong/weak phenotype…”.
8) Another point firmly established by this work that was debated before was whether there is amyloid structure in the M domain. The authors show that there is and that it extends well into M, as was previously suggested by earlier (but still contested) physical and biological studies.
9) This work shows that there are, in a ‘single’ variant (or as close as one can come to that given the ‘cloud’ problem), distinct regions of structure (PK-resistant), separated by PK-sensitive, and therefore presumably largely unstructured or more loosely structured regions. This seemingly disagrees with the previous H-D exchange data which seemed to show just one long continuous slow-exchange region, differing in length between the two variants tested. The current picture is more plausible, and the difference may be explained by the fact that the earlier work was done on in-vitro formed filaments, which may be (must be, in my view) more heterogeneous (represent more variants) than the filaments isolated from in vivo used here.
In summary, this is a really fine paper, with lots of important data that truly moves the ball down the field. The work is clearly presented, with an accurate (!) review of other pertinent literature and relation to the current work. Although I have some minor criticisms, this paper will become a citation classic, as least for me. It is a great tribute to both the authors and their recently deceased mentor, Michael Ter-Avanesyan.
Author Response
Response to Reviewer 2 comments
The authors are very thankful to both Reviewers for good words about our work and the helpful suggestions and corrections. We agree with almost all comments and introduce the following corrections. We cite below only the Reviewer’s text which requires answer.
4) While isolation of filaments from in vivo has been done before, I think this is the first case where the purpose was for examination by MS. I am sure that other groups will want to use this method for examining variant differences that are not defined by ‘strong’ vs ‘weak’. For that reason, the authors should present the filament isolation method, the PK digestion method and the mass spec method in more detail. For example, how was the GFP-fluorescing fraction “collected” (lines 573-574). What kind of tubes were used for the acetone-precipitation of peptides and spinning, all in tiny volumes (lines 576-582)? What problems arose in the method, and how were they dealt with.
Answer: the requested details were added to text. In particular, we indicate that the GFP fraction was collected simply by pipette, the acetone precipitation was made in 0.5 ml PCR tubes, and provide the catalog number for ultracentrifugation tubes. I can comment that our previously published method of prion isolation was substantially modified here, which allowed to substantially improve the quality of spectra. We are also proud of another tiny invention – precipitation of digests with acetone, which is simpler, quicker and more selective than the previously used ultracentrifugation, and it also makes desalting.
5) The authors assert that the [PSI+] variants examined fall nicely into the two classes, ‘strong’ and ‘weak’. But my view of the data suggests that the variants comprise a wide spectrum, with strong and weak representing the two ends in the dimension examined. For example, in Fig. 2A, the color spread seems like a continuous variation from near-white to very dark pink: S-W8 and S-VH (classified as strong) do not seem whiter than W-111 or W-112, all in strain 5V-H19. Likewise S-L6c and S-L20c look the same to me as WS2 in the background of strain 74-D694. While the readthru data in Fig 2B show nice separation of strong from weak groups, it is possible that other terminators or their surrounding sequences may not be so clear. The data in table 1 do show average differences between the strong and weak groups, but I don’t see any fragment that is generally present in one group and not found in the other. Lots of overlap and TREMENDOUS variability within each group.
In fact, a lot of work from many groups, including Kushnirov’s group, shows that yeast prion variants (including [PSI+] variants) can be separated by many different criteria, including lethality (without overproducing anything, many [PSI+] variants are severely toxic or lethal), sensitivity to overproduction or deficiency of various chaperones or other components, ability to overcome a transmission barrier based on Sup35p sequence differences (inter-species, intra-species or artificially created mutants), and other criteria as well. Many of these differences to not follow the strong vs weak paradigm, so it seems clear that the distribution of prion variants is multidimensional.
Answer: This comment does not request corrections, but I wish to counter-comment. We stress in chapter 3.4 that the color phenotype is quite unreliable, as this and other works show, and it can be used only for preliminary classification. The readthrough data are more reliable. Finally, we provide here three criteria, which unambiguously divide all studied [PSI+] variants into two major classes, which in all studied cases coincided with the former approximate division for weak and strong [PSI+]. These criteria are digestion pattern of Core 1 and the effects of multicopy SUP35 and HSP104.
6) Lines 293 to 301 were a bit confusing to me. On first reading, I thought the authors were saying some variants had ONLY region 2 being PK – resistant. I think this is not what they intend, and a bit of re-phrasing might help.
Answer: the text was modified to: “The latter has never been observed for intact Sup35NMG, but it was obtained by overproduction of the Sup35NMG protein lacking the first 30 residues…”
7) Line 318, ”Core 1 mainly defined the phenotype…”. As mentioned in 5) above, there are other phenotypes of [PSI+] that do not correlate with strong vs weak and are not tested here. I suggest the authors make it, “Core 1 mainly defined the strong/weak phenotype…”.
Answer: the text was changed to “Core 1 mainly defined the [PSI+] nonsense-suppressor phenotype..”
Round 2
Reviewer 1 Report
The authors have done a nice job addressing my comments.
Reviewer 2 Report
The paper is ready to publish in my view.